# A Moderate Duration of Stress Promotes Behavioral Adaptation and Spatial Memory in Young C57BL/6J Mice

**DOI:** 10.3390/brainsci12081081

**Published:** 2022-08-15

**Authors:** Lanyan Lin, Jing Zhang, Xiaoman Dai, Nai’an Xiao, Qinyong Ye, Xiaochun Chen

**Affiliations:** 1Department of Geriatrics, Fujian Provincial Hospital, 134 Dongjie Road, Fuzhou 350001, China; 2Fujian Key Laboratory of Molecular Neurology, Institute of Neuroscience, Fujian Medical University, Fuzhou 350005, China; 3Department of Neurology, Fujian Institute of Geriatrics, Fujian Medical University Union Hospital, 29 Xinquan Road, Fuzhou 350001, China

**Keywords:** different stress duration, adaptation, maladaptation, GABAergic-Glutamatergic neuron balances, synaptic plasticity

## Abstract

Stress may serve multiple roles in cerebral functioning, ranging from a highly appropriate behavioral adaptation to a critical risk factor for susceptibility to mood disorder and cognitive impairment. It is well known that E/I (excitation/inhibition) balance is essential for maintaining brain homeostasis. However, it remains largely unknown how GABAergic and Glutamatergic neurons respond to different stressful stimuli and whether the GABAergic-Glutamatergic neuron balance is related to the transition between adaptive and maladaptive behaviors. Here, we subjected 3-month-old mice to chronic mild stress (CMS) for a period of one, two, and four weeks, respectively. The results showed that the two-week CMS procedure produced adaptive effects on behaviors and cognitive performance, with a higher number of GABAergic neuron and VGluT1-positive neurons, increasing the expressions of p-GluN2B, Reelin, and syn-PSD-95 protein in the hippocampus. In contrast, the prolonged behavioral challenge (4 week) imposes a passive coping behavioral strategy and cognitive impairment, decreased the number of GABAergic neuron, hyperactivity of VGluT1-positive neuron, increased the ratio of p-GluN2B, and decreased the expression of Reelin, syn-PSD-95 in the hippocampus. These findings suggest that a moderate duration of stress probably promotes behavioral adaptation and spatial memory by maintaining a GABAergic-Glutamatergic neuron balance and promoting the expression of synaptic plasticity-related proteins in the brain.

## 1. Introduction

Emotional and stressful experiences are commonplace in our daily lives. Compared with events of low emotional value, stressful incidences are more likely to be remembered [1]. This memory, from an evolutionary perspective, may be highly adaptive as it facilitates coping strategies for similar situations in the future [2]. Studies have evidenced that the exposure to such stressful experiences can initiate adaptive processes, which allow organisms to physiologically cope with prolonged or intermittent exposure to real or perceived threats and that the initiated behavioral adaptation is mediated by ongoing physiological responses [3,4]. Adaptive and favorable as it may seem initially, prolonged stress exposure may instead induce pronounced physiological and behavioral changes that have long-term detrimental effects on the survival and well-being of an individual [2,5,6]. In the extreme, this may contribute to the development of psychiatric conditions, such as general anxiety disorder, major depressive disorder, or post-traumatic stress disorder [7]. Therefore, it is crucial and significant to explore how stress regulates brain function and what determines the threshold between adaptive and maladaptive responses.

In the available literature, the excitatory/inhibitory (E/I) balance allows both directionally and recurrently wired networks to promote information processing while preventing runaway activity [8]. Moreover, disruption of the E/I balance has been linked to a number of neuropsychiatric disorders, including schizophrenia, autism, and epilepsy [8,9,10]. In normal neurologic function, it is crucial to maintain a balance between glutamate, the primary excitatory neurotransmitter, and γ-aminobutyric acid (GABA), the principal inhibitory neurotransmitter [11]. In stressful conditions, such as severe epileptic seizures or stroke, overactivation of glutamate receptors can kill neurons during excitotoxic response coupled with calcium influx [12]. However, the subtoxic levels of the activated N-methyl-Daspartate (NMDA) type of glutamate receptors may elicit adaptive responses in neurons so as to enhance their ability to withstand severer stress and to protect neurons [13,14,15]. Current evidence demonstrates that the environmental stress or artificially elevated glucocorticoid levels can upregulate the GABAergic elements, resulting in changes in the GABAergic system (the main inhibitory system in the brain) [16,17]. However, it remains largely unknown how GABAergic neurons and Glutamatergic neurons respond to these different stressful stimuli and whether the GABAergic-Glutamatergic neuron balance is related to the transition between adaptive and maladaptive behaviors.

In the current study, we subjected 3-month-old mice to chronic mild stress (CMS) for a period of one, two, and four weeks, respectively. The results showed that the two-week eustress produced adaptive effects on behaviors and cognitive performance by maintaining a balance between GABAergic neurons and Glutamatergic neurons, promoting the synaptic-plasticity-related proteins in the brain, while the prolonged behavioral challenge (4-week stress) imposed a passive coping behavioral strategy and cognitive impairment by disrupting the balance between GABAergic neurons and Glutamatergic neurons and reducing the levels of synaptic-plasticity-related proteins in the hippocampus. These findings may provide novel insights into the underlying mechanisms for the adaptive-to-maladaptive response switch.

## 2. Materials and Methods

### 2.1. Animals and Experimental Protocol

Young male C57BL/6J mice (aged 12 weeks, *n* = 60) were randomly divided into two groups: chronic mild stress group and CMS group (*n* = 30), which were randomly divided into three subgroups (*n* = 10 per subgroup); and those kept under normal conditions, control group (*n* = 30), which were also randomly divided into 3 subgroups (*n* = 10 per subgroup). All experimental protocols were approved by the Institutional Animal Care and Utilization Committee of Fujian Medical University (the ethical approval code of experimental animals is ID: FJMU IACUC 2021-0399) and closely observed the National Institutes of Health’s “Guide for the Care and Use of Laboratory Animals” (NIH Publication 80-23).

### 2.2. Chronic Mild Stress (CMS) Procedure

The chronic mild stress (CMS) process was described in our previous study, adapted from Willner et al., with minor modifications [18,19,20]. For the CMS groups, each mouse was isolated in a separate cage and subjected to a variety of mild stressors within one week: white noise for 2 h (80 dB), cage tilt at 45° for 4 h, swimming in 18 °C water for 5 min, water deprivation for 6 h, food deprivation for 6 h, strobe lights for 2 h (300 flashes/min), and restrictions in a small tube (10 × 5 × 5 cm) for 30 min. The same stress procedure lasted for one, two, and four weeks, respectively. After the intervention, the animals received weight and sucrose solution consumption tests. The behavioral tests were carried out in the sequence of OFT (Open Field Test), EPM (Elevated Plus Maze), TST (Tail Suspension Test), and MWM (Morris Water Maze). After all behavioral tests, all mice were sacrificed for further molecular biological detection. The detailed schedule is shown in Figure 1A,B.

### 2.3. Sucrose Solution Consumption Test (SSC)

The sucrose preference training was conducted before the experiment and sucrose solution consumption test was performed after the intervention. The detailed protocol is as follows: the sucrose preference test was conducted from 10:00 a.m. on the starting day to 10:00 a.m. on the next day. The animals had free access to two bottles containing water and 1% sucrose solution. The consumption of water and sucrose solution (the percentage of sucrose preference) was measured the next day.

### 2.4. Behavioral Testing

#### 2.4.1. Open Field Test (OFT)

The open field (OFT) was designed following the procedure described by Heimrich et al. [21]. Each mouse was placed in the middle of an open box (50 × 50 × 50 cm) and the area was divided into 9 squares (16.67 cm × 16.67 cm per square) and painted with white line. During the 10 min test, mouse activity was recorded by a digital video camera (SONY, Japan) and analyzed by Top Scan software (Super Maze V2.0, XinRuan Information Technology Co. Ltd., Quanzhou, China). Finally, these parameters including the numbers of those crossing the central grid, times of climbing and standing, total numbers of crossing grids were analyzed.

#### 2.4.2. Elevated Plus Maze Test (EPM)

The Elevated Plus Maze (EPM) was described by Pellow et al. [22]. The maze device (50 cm above the ground) consists of two open arms (50 × 5 cm; with 0.5 × 0.5 cm ledges) and two closed arms (50 × 5 cm; with 38 cm high walls) and connected by a 5 × 5 cm central area. Each mouse was individually placed in the central area and allowed to explore freely for 10 min. Then these parameters including the ratio of time entering open arm and the ratio of number entering open arm were calculated.

#### 2.4.3. Tail Suspension Test (TST)

The tail suspension test (TST) was conducted according to Crowley et al. [23]. The mice’s tail was tied with a piece of tape and fixed on a hook. After 2 min acclimation, the percentage of resting time for each mouse was recorded by an automated device (Super Maze V2.0, Xinruan, China) for a period of 6 min.

#### 2.4.4. Morris Water Maze (MWM) 

Morris Water Maze (MWM) [24] was adopted for the evaluation of spatial cognitive performance in the mice. The apparatus and procedure of the water maze are described in our previous study [20]. In brief, in the spatial cognitive test, each mouse was tested in four trials daily for five consecutive days, starting at a different position determined by semi-random sequence distribution [25] in each trial (each trial is 60 s). For the sixth day, after removing the platform, each mouse was allowed to freely explore the pool within 60 s. All performance was recorded by a Smart 2.0 video tracking software (PanLab, Barcelona, Spain). At the end of the experiment, these parameters including escape latency, the percentage of time target quadrant, and the numbers of crossing were analyzed.

### 2.5. Tissue Preparation

As previously described [20], all animals were anesthetized with 10% chloral hydrate and perfused with 0.1 M PBS (about 25 mL per mouse) from the left ventricle. Then the brain tissue was quickly removed from the skull and immediately placed on ice to isolate the hippocampus. Some of the hippocampus was immediately immersed in liquid nitrogen and stored at −80 °C. Some hippocampal tissues were fixed in 4% paraformaldehyde for 2 days at 4 °C and dehydrated twice by 30% sucrose solution (24 h each time). The fixed tissues were cut into 30 um thick serial sections with a freezing microtome (CM1850, Leica) and stored at −20 °C in antifreeze solution.

### 2.6. Western Blot Analysis

The protein concentration in the tissues was determined with the Bradford assay kit (Bio-Rad, Hercules, CA, USA). The total and synaptic (Syn) protein extraction (Syn-PER, Thermo Scientific, Waltham, MA, USA) was performed as previously described [19]. The protein lysates were separated by SDS-PAGE (about 10–12%, sodium dodecyl sulfate, polyacrylamide gel electrophoresis) and transferred to PVDF (polyvinylidene fluoride) membranes. The membranes were blocked in 5% BSA (bovine serum albumin), then incubated with primary antibodies at 4 °C overnight, next incubated off light at RT (room temperature) in secondary antibodies for 1 h (IRD 800 cw, goat-rabbit or goat-mouse 1:10,000; LI-COR). Then, the fluorescence was detected by Odyssey Sa image system (LI-COR) and the densitometric readings were analyzed by Image J software. All antibodies used are as follows: mouse anti-β-actin (1:50,000, Sigma), mouse anti-tubulin (1:50,000, Sigma, Darmstadt, Germany), mouse anti-Reelin (1:500, Milipore, Burlington, MA, USA), rabbit anti-PSD-95 (postsynaptic density protein) (1:2000, Millipore), and rabbit anti-GluN2B, anti-GluN2A, and anti-phosphorylation-GluN2B,(1:1000, Abcam, Cambridge, UK), respectively.

### 2.7. Immunohistochemistry

The sections were blocked in the solution (containing 0.3% Triton X-100, 0.25% BSA (bovine serum albumin) and 5% GS (goat serum)) for 2 h at RT. Next, they were incubated with primary antibodies (anti-GABA, 1:8000, Sigma) overnight at 4 °C, then incubated with secondary antibodies (1:6000; Vector Laboratories, Burlingame, CA, USA) for 90 min at RT. Subsequently, the sections were incubated with Vector Elite affinity peroxidase dilution (1:200) for 60 min at RT. Afterwards, the sections were incubated with DAB (diaminobenzidine) for 10 min at RT and mounted on slides.

For anti-GABA-positive cells, cell counting was performed as previously described [19]; the sections were observed by an OlympusBX-51 microscope (Olympus, Tokyo, Japan). Image-pro Plus Image analysis software was used for image analysis. For quantitative analysis, the serial section technique was adopted (one every 5 sections was selected) and total of 5 sections was measured for each mouse. The dentate gyrus (DG) was selected as ROI (region of interest) and the numbers of positively stained neuron were counted by magnification of 100 times. The “cells” with clear brown boundaries were counted and the “cells” with lighter staining or irregular shape would be excluded from quantification. The number of positive neurons per square millimeter was calculated for each mouse.

### 2.8. Immunofluorescence

For immunodetection of Glutamatergic neurons, the sections were blocked in the solution (containing 0.3% Triton X-100, 0.25% BSA (bovine serum albumin) and 5% GS (goat serum)) for 2 h at RT. Then the sections were incubated with primary antibodies (rabbit anti-vGluT1, 1:1000, Abcam; rabbit anti-NeuN, 1:3000, Abcam) overnight at 4 °C and with fluorescent secondary antibody (at 1:1000, Abcam) at RT for 90 min. After DAPI incubation for 5 min, they were mounted by an anti-fluorescence quencher. Finally, images of the sections were taken with a confocal fluorescence microscope (Leica) and fluorescence intensity was quantified with the Image-Pro plus software. Image J software was used to digitize the fluorescence images and the figures were converted into 8-bit gray images. Then the selected area was analyzed to determine its surface area and the average, minimum, and maximum gray values and integrated optical density (IOD) were calculated. All images were set to the same light intensity and exposure. The sample size for each group was *n* = 5.

### 2.9. Real-Time Polymerase Chain Reaction (RT-PCR)

Total RNA was extracted by TriPure isolation reagent (Roche) and reverse transcribed by cDNA Synthesis Kit (Ferments). Then PCR was performed by Universal SYBR Green Master (Roche). Finally, the fluorescence assay was performed by a real-time PCR system (Grand Island). The primer sets were used as follows: Reelin (NM_011261) and GAPDH (NM_008084).

### 2.10. Statistical Analysis

All data were analyzed by SPSS 13.0 statistical software and mean ±SEM are used to express quantitative data. The data sets in Figure 1H (Escape latency) were tested for normal distribution firstly and analyzed by Mauchly’s test of sphericity and the two-way Repeated Measures(RM)analysis of variance (ANOVA). For the other figures, the data were also tested for normal distribution and assessed by two-way ANOVA and Tukey post hoc test. Statistical significance was set at *p* < 0.05.

## 3. Results

### 3.1. A Moderate Duration of Stress Induces Adaptive Effects on Behavioral State and Spatial Memory

To assess the effect of different stress durations, the body weight and sucrose preference were measured undergoing the CMS for one, two, and four weeks, respectively. The results showed that the body weight and sucrose consumption preference of mice undergoing the one-week and two-week CMS procedures were not significantly different with the control groups, while those in the four-week CMS group were greatly lower than those in the control group (body weight: 22.51 ± 0.32 g vs. 24.69 ± 0.20 g, Tukey post hoc *p* < 0.01; sucrose consumption preference: 65.60 ± 2.47% vs. 77.10 ± 2.03%, Tukey post hoc *p* < 0.01) (Figure 1C,D). These data suggest that a moderate amount of stress may produce beneficial effects and help the mice to resist environmental stressors, while a long-term chronic stress may exert harmful effects on the health of the mice.

To evaluate the impacts of different stress durations on the behaviors of mood disorders, including depression and anxiety in mice, the open field test, elevated plus maze, and tail suspension test were employed in sequence. From the tail suspension test, it revealed that the four-week CMS group performed a depression-like state, whose percentage of resting time was longer than that in the control groups (57.22 ± 3.30% vs. 43.10 ± 2.22%, Tukey post hoc *p* < 0.01) (Figure 1E), while that in the two-week CMS group was much shorter than that in control group (36.25 ± 1.15% vs. 43.55 ± 1.38%, Tukey post hoc *p* < 0.01) (Figure 1E). We also used the open test and elevated plus maze to detect anxiety-like behaviors in mice. From the open field test, compared with the control group, the number of crossings over the central grid in the two-week CMS mice significantly increased (82.20 ± 1.95 times vs. 70.90 ± 2.52 times, Tukey post hoc *p* < 0.01) (Figure 1F), while that in the four-week CMS group greatly decreased (59.00 ± 2.69 times vs. 71.70 ± 2.21 times, Tukey post hoc *p* < 0.01) (Figure 1F); and that in the one-week CMS group was not significantly different from its control group (67.70 ± 2.70 times vs. 73.40 ± 2.15 times, Tukey post hoc *p* > 0.05) (Figure 1F). In terms of grooming and climbing, the one- and two-week CMS groups were not so different from the control groups (Tukey post hoc *p* > 0.05) (Figure 1F), while the four-week CMS mice performed less grooming and climbing when compared with the control mice (39.10 ± 1.53 times vs. 45.70 ± 1.33 times, Tukey post hoc *p* < 0.01) (Figure 1F). For the total movement, no significant difference was found across all the groups (Figure 1F). Furthermore, compared to the control group, the four-week CMS group spent significantly less time on the open arms of the elevated plus maze (41.29 ± 1.91% vs. 58.80 ± 2.90%, Tukey post hoc *p* < 0.001) (Figure 1G) and less frequently entered into the open arms undergoing four weeks of CMS procedure (38.30 ± 1.96% vs. 51.65 ± 2.27%, Tukey post hoc *p* < 0.001) (Figure 1G). Notably, compared with the control mice, the two-week CMS mice displayed more interest in the open arms, in which they spent more time and entered more frequently into the open arms of the elevated plus maze (for the time spent in the open arms: 66.06 ± 1.90% vs. 55.02 ± 2.62%, Tukey post hoc *p* < 0.01; for the number of entries into the open arms: 59.99 ± 2.51% vs. 48.38 ± 1.25%, Tukey post hoc *p* < 0.01) (Figure 1G); however, no obvious difference was found in the one-week group. Taken together, these data suggest that the two-week CMS procedure produces adaptive effects on the behavioral state, while the four-week stress imposes a passive coping behavioral strategy, including depression- and anxiety-like behaviors.

The hippocampus is a medial temporal lobe structure that plays an important role in declarative memory in humans [26,27] and spatial working memory in rodents [28,29]. We adopted the Morris Water maze to detect hippocampal-dependent spatial memory. Regardless of the training time, there was no significant change in escape latency in the four-week CMS group (DAY 4, 40.29 ± 1.20 s vs. 24.69 ± 1.98 s, two-way RM ANOVA *p* < 0.001; DAY 5, 36.56 ± 3.09 s vs. 17.43 ± 2.78 s, two-way RM ANOVA *p* < 0.05) (Figure 1H). Noteworthily, the two-week CMS group displayed a shorter escape latency when compared with that of control group (DAY 5, 10.97 ± 0.76 s vs. 17.40 ± 2.69 s, two-way RM ANOVA *p* < 0.05) (Figure 1H). In the probe trial, removing the platform, the four-week CMS group spent less time exploring the target quadrant (38.95 ± 2.46% vs. 56.85 ± 3.11%, Tukey post hoc *p* < 0.01) (Figure 1H). Additionally, in the four-week CMS mice, the numbers crossing the platform position were lower than those in the control group (2.20 ± 0.25 times vs. 5.0 ± 0.52 times, Tukey post hoc *p* < 0.001) (Figure 1H). Furthermore, compared with the control group, the two-week CMS mice spent much longer time in the target quadrant (74.17 ± 1.96% vs. 58.70 ± 3.47%, Tukey post hoc *p* < 0.01) (Figure 1H) and more often crossed over the platform position (6.80 ± 0.33 times vs. 5.10 ± 0.46 times, Tukey post hoc *p* < 0.01) (Figure 1H). These data indicated that long-term chronic stress may impair the spatial memory, while a moderate duration of stress may help maintain a good spatial memory in a CMS confrontation.

### 3.2. A Moderate Duration of Stress Helps Maintain the GABAergic-Glutamatergic Neuron Balance in the Hippocampus

The dentate gyrus (DG) in the hippocampus is the residence of neural stem cells (NSC), which produces neural progenitors that can differentiate into neural lineage, which is essential for the generation of new granule neurons [30,31,32]. Abnormalities in neuronal function in the DG affect the entire neurogenesis process and impaired contextual fear conditioning, memory function, and anxiety- and depression-like behaviors [33,34].

GABAergic neurons are widely distributed in the DG region of the hippocampus [35]. In order to assess the expression of GABAergic neurons in the controls and CMS groups, the expressions of GABA were examined by immunohistochemistry. As shown in Figure 2, the number of anti-GABA-positive neurons in the hippocampus of the four-week stress group was lower than that in the control group (58.52 ± 2.98 cells/mm^2^ vs. 79.44 ± 1.72 cells/mm^2^, Tukey post hoc *p* < 0.01) (Figure 2A,B); however, after a variety of eustress, that in the two-week stress group was increased when compared with that in the control group (88.18 ± 1.38 cells/mm^2^ vs. 76.84 ± 1.93 cells/mm^2^, Tukey post hoc *p* < 0.01) (Figure 2A,B) and no obvious difference was present between the one-week-stress mice and the control group (77.30 ± 2.46 cells/mm^2^ vs. 81.60 ± 2.61 cells/mm^2^, Tukey post hoc *p* > 0.05) (Figure 2A,B). These data suggest that long-term CMS intervention decreases the number of GABAergic neurons in the hippocampus of mice, while a moderate stimulus may upregulate the number of the GABAergic neurons in the hippocampus.

Apart from the GABAergic neurons in the DG of the hippocampus, the Glutamatergic neurons are also closely related with depression and cognitive impairment [11], so we detected the immunofluorescence of VGluT1(the presynaptic gutamatergic transporter), a significant Glutamatergic neuron marker, in the DG of the hippocampus [36]. The VGluT1-positive neurons were quantified by adapting the semi-quantitative method (IOD measurement). From Figure 3A, the fluorescence intensity of VGluT1-positive neurons in the two-week CMS group was slightly higher than the control group (0.166 ± 0.011 IOD/mm^2^ vs. 0.132 ± 0.0089 IOD/mm^2^, Tukey post hoc *p* < 0.05) (Figure 3A,B). However, in the four-week CMS group, the mean fluorescence intensity of VGluT1-positive neurons was greatly increased compared with that in the control group (0.194 ± 0.011 IOD/mm^2^ vs. 0.136 ± 0.0073 IOD/mm^2^, Tukey post hoc *p* < 0.01) (Figure 3A,B). No obvious difference in the fluorescence intensity of vGluT1 in the hippocampal region was found between the one-week CMS group and the control group (Tukey post hoc *p* > 0.05) (Figure 3A,B). We also detected the expression of the postsynaptic receptor NMDAR, an important Glutamatergic neuron marker. As shown in Figure 3C, the level of p-GluN2B was elevated in the two-week CMS group when compared with that in the control group (40.58 ± 3.56% increase, Tukey post hoc *p* < 0.01) (Figure 3C,D), while that in the four-week CMS group excessively increased when compared with that in the control group (52.30 ± 1.90% increase, Tukey post hoc *p* < 0.001) (Figure 3C,D). Moreover, the expressions of GluN2A and GluN2B were not significantly different from each other across all the groups (Tukey post hoc *p* > 0.05) (Figure 3C,D). Altogether, our data suggest that a moderate duration of stress may produce a light activation of Glutamatergic neurons in the hippocampus of mice, while a long-term CMS intervention may trigger their overactivation.

### 3.3. A Moderate Duration of Stress Facilitates the Expression of Synaptic-Plasticity-Related Proteins Reelin and Syn-PSD-95 in the Hippocampus of Mice

Given that the signaling pathway of Reelin and PSD-95 plays an important role in emotional disorders and cognition [19], the expressions of Reelin and PSD-95 in the hippocampus were examined by Western blotting. As shown in Figure 4C, in the hippocampus, the level of Reelin (molecular weight 170 Kda) in the four-week stress group was much lower, relative to that in the control group (29.73 ± 2.99% decline, Tukey post hoc *p* < 0.01), while that in the two-week stress group was higher than the control group (41.48 ± 6.09% increase, Tukey post hoc *p* < 0.01) (Figure 4C). The mRNA level of Reelin in the hippocampus was also detected, reflecting a trend similar to that of the protein, with the mRNA expression of Reelin declining in the four-week stress group (49.35 ± 3.41% decline, Tukey post hoc *p* < 0.001) and increasing in the two-week stress group (36.20 ± 2.70% increase, Tukey post hoc *p* < 0.01) (Figure 4B). The expression of PSD95 in the synaptic of the hippocampus in the two-week stress group increased when compared with that in the control group (30.52 ± 2.68% increase, Tukey post hoc *p* < 0.01) (Figure 4C), while that in the four-week stress group was much lower than that in the control mice (34.85 ± 2.60% decline, Tukey post hoc *p* < 0.01) (Figure 4C). However, the level of total-PSD-95 was not significantly different across all the groups (Tukey post hoc *p* > 0.05) (Figure 4C). These results indicate that a moderate duration of stress upregulates the amount of Reelin and syn-PSD-95 in the hippocampus of mice, which is crucial for the maintenance of synaptic plasticity.

## 4. Discussion

In the present study, mice undergoing a two-week stress procedure displayed behavioral adaptation and cognitive performance, an increase in GABAergic neurons, and a moderate activation of VGluT1-positive neurons, which, in turn, upgraded the level of p-GluN2B, Reelin, and syn-PSD-95 in the hippocampus. In contrast, the prolonged behavioral challenge (4-week stress) imposed a passive coping behavioral strategy and cognitive impairment, decreased the number of GABAergic neurons, hyperactivity of VGluT1-positive neurons, and increased the ratio of p-GluN2B, as well as decreasing the expression of Reelin and syn-PSD-95 in the hippocampus.

Stress is often thought as highly heterogeneous and variable experience, depending on factors, such as time point upon exposure, duration of exposure, severity, controllability, and predictability [37,38]. The dual role of stress in learning and memory is influenced by the duration and intensity of the stressors [39]. Some studies have shown an inverted “U” relationship between stress and cognitive function, suggesting that moderate increases in the levels of corticosterone (mild stress) may lead to pro-cognitive effects, whereas large increases in corticosterone levels (chronic stress) may adversely affect cognitive processing [40,41,42]. Available evidence suggests that instances of eustress may actually promote neurobiological adaptations [43,44].

Consistent with these findings, in the current study, 3-month-old mice, receiving two-week mild stress intervention, demonstrated adaptive behaviors when compared with the control mice, as shown in the normal weight growth, greater sugar water preference, more crossings over the center grid in the open field, more entries into the open arm (Figure 1C,D,F), and enhanced spatial memory in the Morris maze test (Figure 1H). These results may evidence adaptive effects and improved performance in cognitive tasks. However the long duration of restraint stress (about 28 days) reveals a maladaptive state with a gradual loss of body weight and sucrose preference, drastic reductions in crossings over the center grid, self-grooming, climbing in the open field, entries into the open arm in the elevated maze (Figure 1C,D,F), and impaired spatial memory in the Morris Water maze test (Figure 1H). Yet, no significant difference was found in the one-week group for all the above tests. Our study suggests that the two-week mild stress procedure may facilitate the adaptability of the mice to the eustress.

Some studies indicate that moderate beneficial stress induces elevated cortisol levels, which lead to cravings for sweets, fatty food, and increased exploratory activity, while long-term variable stress (over 3 weeks), due to prolonged and repeated increases in cortisol, glucocorticoid receptors (GR), inflammatory cytokines, etc., evoked neuroendocrine response, resulting in decreased food intake or even anorexia, weight loss, and emotional disorders, such as prolonging immobility time and decreasing desire to explore. Our results are consistent with the changes in cortisol in the mice exposed to stress [45,46,47]. In our study, we set the same intensity and type of stress, with an extension of intervention time, the 4-week CMS mice showed weight loss and reduced sugar water consumption. In the behavioral tests, the 4-week CMS mice had prolonged immobility time and decreased desire to explore, such as reducing access to the central zone and open arms, and performed poorly in spatial memory, while the 2-week CMS mice supplemented the energy consumed by increasing the craving for sweets (increased sugar water consumption) and their body weight did not decrease significantly. In behavioral tests, mice showed a strong desire to explore, such as decreasing immobility time in TST, increasing activities in the central area and open arm, and showing excellent spatial cognition in MWM. It is indicated that different durations of stress may trigger biphasic effects on emotional stability hippocampal excitability and cognitive function through neuroendocrine responses. The short-term stress may contribute to cortisol-induced adaptive behavior in response to eustress. However, excessive and sustained stress can have serious maladaptive effects.

The balance between excitatory and inhibitory neurotransmission in the brain is essential for performing a range of higher-order functions, such as planning, working memory, decision making, emotional regulation, and error monitoring [48]. GABAergic interneurons, as the only inhibitory interneurons in the brain, are the main regulator of E/I balance. GABAergic neurotransmission is highly plastic and dynamically affected by environmental changes and stress-related GABAergic inhibition contributes to the alteration in neuronal excitability [49]. Some studies reported that impairments in GABAergic neurotransmission in the limbic system are associated with major depression [50,51,52]. Our data showed that the 2-week CMS intervention upregulated the numbers of the GABAergic neurons and light activation of VGluT1-positive neurons and that the 4-week CMS group displayed a decrease in the GABAergic neurons and an excessive overload of Glutamatergic neurons in the hippocampus, with a major difference in the number of the GABAergic neurons between the two CMS groups, which is consistent with the dynamic adaptive-maladaptive shift in mouse behaviors. These findings suggest that the changes in the number of inhibitory GABAergic neurons may account for the behavioral transformation observed in those mice subjected to chronic stress. We speculated that GABAergic neurons, the main inhibitory neurons, may play an important role in regulating the activation of Glutamatergic neurons. The long-term stress may significantly weaken the inhibitory effect of GABAergic neurons on the excitability of Glutamatergic neurons, resulting in maladaptive behaviors. Conversely, eustress may promote the inhibitory effect of GABAergic neurons, limiting the excitotoxic effects of Glutamatergic neurons, so as to develop adaptive behaviors.

In depression models, chronic stress can disrupt plasticity, foster neuronal atrophy, reduce synaptic number and function, especially in the hippocampus, and result in maladaptation to environment, impairing stress or learning coping [53]. Conversely, when neuroplasticity is enhanced (for example, by treatment), synaptic contacts increase, strengthening the adaptability by allowing active-dependent competition to stabilize neural structures [54]. Some studies suggest that antidepressant treatment may increase hippocampal volume through the generation of new neurons and specific structural changes in the dendrites of existing neurons [55,56,57]. In reconciling these findings, the heterogeneity of stressful experiences should be the focus of attention. Our study found that a moderate stress duration of about two weeks was beneficial to increase the expression of Reelin and the aggregation of syn-PSD-95 protein in the synapses. On the contrary, the long-term continuous stress (over 4 weeks) greatly reduced the expression of Reelin protein in the hippocampus, along with the chronic-stress-induced downregulation of the syn-PSD-95 protein in the synapses. Therefore, it indicated that decreased expression of Reelin and aggregation of PSD-95 protein in the synapses may be linked to one of the mechanisms underlying the transition from adaptation to pathology [19,20,58,59,60].

In conclusion, our study confirms that a duration of appropriate stress intervention (2-week stress) probably promotes behavioral adaptation and spatial memory in mice, while a prolonged behavioral challenge (4-week stress) imposes a passive coping behavioral strategy and cognitive impairment, which provides insights into the potential mechanisms underlying the transition from adaptive and maladaptive responses in young mice. In part, the effects of stress are caused by the corticosterone (CORT), produced by the adrenal cortex in response to stress, due to the abundant expression of the glucocorticoid receptor (GR) in the hippocampus, which is particularly sensitive to corticosteroids [61]. Some studies found that chronic-stress-induced elevation of CORT can slowly produce excitatory axis-spinous synaptic connection neuronal cell damage in the CA1 layer of rats [62]. Unfortunately, the limitation of our work is that we did not detect physiological markers (such as cortisol, GR, and inflammatory cytokines in mice) for different durations of stress. Our future research will focus on the role of stress-cortisol-hippocampus (stratum pyramidal) in adaptive and maladaptive behavioral transitions.

## Figures and Tables

**Figure 1 brainsci-12-01081-f001:**
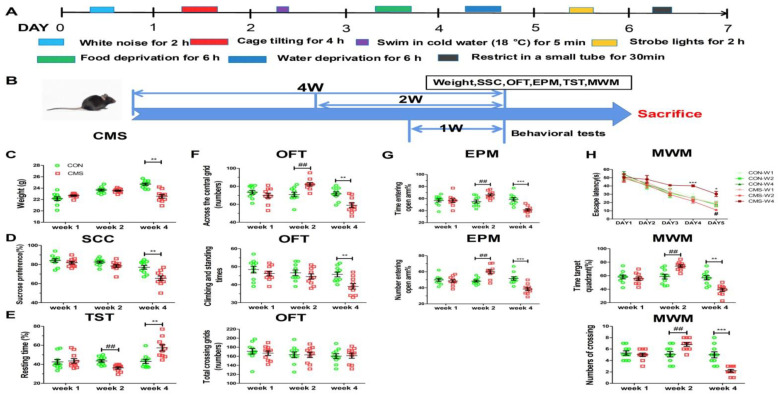
The behavioral adaptation and the improved spatial memory in C57BL/6J mice by a moderate duration of stress. (**A**) The detailed CMS procedure of 3-month-old C57BL/6J mice for each day. (**B**) The time schedule for experimental procedures in C57BL/6J mice. (**C**) Body-weight changes in the one-, two-, four-week CMS groups and controls. (**D**) The percentages of sucrose preference in mice undergoing one, two, and four weeks of CMS, respectively. (**E**) The percentage of resting time of 3-month-old mice during the 6 min test. (**F**) The numbers of crossings over the central grid, climbing and standing, total movement for all groups in OFT. (**G**) The percentage of time and numbers of entering into the open arms for the 3-month-old mice in EPM. (**H**) Escape latency in all mice for the first five days in MWM. The percentage of time spent in target quadrant and the numbers of crossing over the former platform position on the sixth day of MWM. N = 10 per group, each sample was not duplicated, expressed as mean ±SEM. * *p* < 0.05, ** *p* < 0.01, *** *p* < 0.001, 4-week CMS vs. 4-week CON; ^##^
*p* < 0.01, 2-week CMS vs. 2-week CON.

**Figure 2 brainsci-12-01081-f002:**
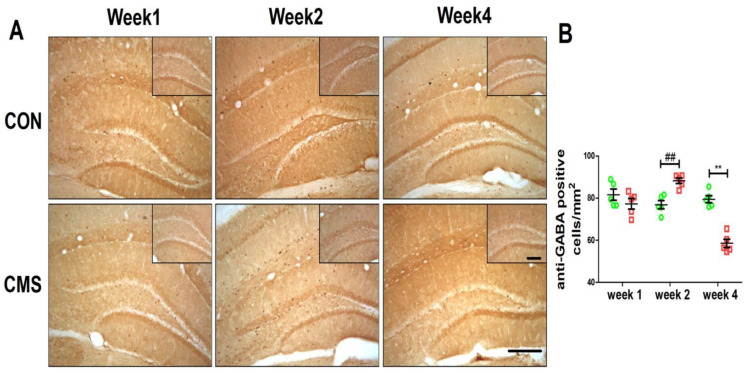
The upregulation of GABAergic neurons in the hippocampus by a moderate duration of stress. (**A**) Anti-GABA immunohistochemical staining of hippocampal tissue of 3-month-old mice. The dark brown dots were considered as anti-GABA-positive GABAergic neurons. (**B**) Quantitative analysis: anti-GABA-positive cells per square millimeter. The numbers of mice were N = 5 per group and the numbers of sections used N = 5 per mouse; the data were expressed as mean ± SEM. ** *p* < 0.01, 4-week CMS vs. 4-week CON; ^##^
*p* < 0.01, 2-week CMS vs. 2-week CON. Scale bars are 200 um.

**Figure 3 brainsci-12-01081-f003:**
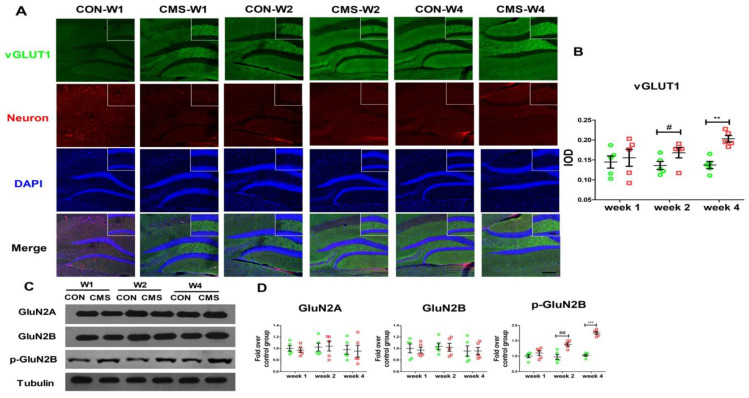
The limited activation of the Glutamatergic neurons in the hippocampus of mice by a moderate duration of stress. (**A**,**B**) Immunofluorescence of VGluT1 in the DG of hippocampus in the 3-month-old mice. VGluT1 (Green), Neuron (Red), DAPI (Blue), and the merge pictures (×10 magnification) were presented on the left side. Quantification of the result (IOD) was shown on the downside. The numbers of mice were N = 5 per group and the numbers of sections used N = 5 per mouse. (**C**,**D**) The levels of the postsynaptic receptors GluN2A, GluN2B, and p-GluN2B were detected with Western blot in the hippocampus, respectively. N = 5 per group, each sample was repeated three times, expressed as mean ± SEM. ** *p* < 0.01, *** *p* < 0.001, 4-week CMS vs. 4-week CON; ^#^
*p* < 0.05, ^##^
*p* < 0.01, 2-week CMS vs. 2-week CON. Scale bars are 200 um.

**Figure 4 brainsci-12-01081-f004:**
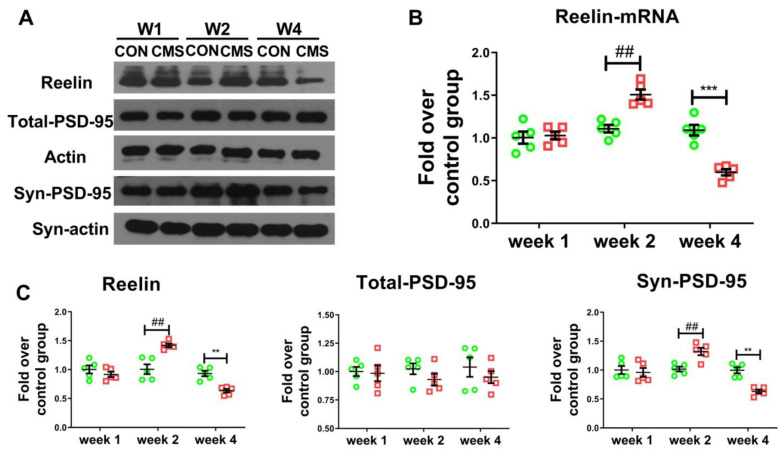
Increased expressions of Reelin and syn-PSD-95 in the hippocampus of mice by a moderate duration of stress. (**A**,**C**) The levels of Reelin (170Kd), Total-PSD-95, Syn-PSD-95 were detected, respectively, in the hippocampus. (**B**) The expression of Reelin-mRNA was detected by RT-PCR. N = 5 per group, each sample was repeated three times, expressed as mean ± SEM. ** *p* < 0.01, *** *p* < 0.001, 4-week CMS vs. 4-week CON; ^##^
*p* < 0.01, 2-week CMS vs. 2-week CON. Synaptic (syn).

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
