# Peer review of "A Moderate Duration of Stress Promotes Behavioral Adaptation and Spatial Memory in Young C57BL/6J Mice"

_brainsci, 2022, doi:10.3390/brainsci12081081_

Round 1
Reviewer 1 Report
In this manuscript, authors have studied the effect of chronic mild stress (CMS) on depression-like behavior and spatial memory. They exposed 3 months old mice to CMS for 1,2 and 4 weeks. They have concluded that moderate amount of CMS exposure for 2 weeks promotes behavioral adaptation and spatial memory enhancement which was supported by behavioral tests such as open field test, elevated plus maze, Morris-water maze, sucrose preference test, tail-suspension test. These behavioral outcomes were supported by immunohistochemical and western blot evaluation of memory related protein markers. They found that chronic 4-week CMS exposure causes increased depression-like behavior as evident by reduced sucrose intake and increased resting time on tail suspension test, spending more time on corners of the open field,while they found opposite results in mice exposed to 2 weeks. 2-week exposed group displaced enhanced spatial memory as they displayed shorter time to find the platform on Morris water maze after 4th and 5th trial. They also found GABAergic neurons in DG were increased in 2-weeks while decreased in 4-weeks CMS exposed groups., VGlut1 positive excitatory neurons were significantly increased in 4-weeks exposed groups, phosphorylated-GluN2B were also significantly increased in 4 weeks exposed group which were moderately increased in 2-weeks CMS group.They also found increased expression of Reelin and synaptic-PSD95 in 2-weeks CMS exposed group while they were reduced in 4-weeks exposed groups.
This is a valuable finding. I only have some minor comments. 1. One of my main concerns is the way it is written. Most of the words used for example in the abstract are non scientific. It looks like computer generated words. Authors should modify it significantly. If you see in the abstract, there is inappropriate used -respectively. There are words which are not common in science writing like obviously, promoted, imposes, excessively up-regulated, boosted, limiting.....so on...so, rewrite the text in a scientific way. 2. I suggest writing the name of the statistical test before the p-value for each result. 3. Morris water maze (MWM) is a more commonly used name than Morris-maze test (MMT). 4. In figure 2, adding a magnified region of DG would be helpful. Select a region in DG and add a magnified image embedded in the same image. 5. In figure 3A, what is the neuronal marker? Is there a specific antibody you used? it's not mentioned in the method section as well. I guess DAPI should be the same across all groups. However it looks faint on the CON-W1 group. Also, 3E is not necessary. 6. line 337, how did you calculate the activating rate of p-GluN2B? from the exp you performed, its not possible to calculate the activity. Probably mentioning phosphorylated GluN2B or activated GluN2B might be appropriate.Author Response
请参阅附件。

Reviewer 2 Report
In the manuscript entitled “A moderate duration of stress promotes behavioral adaptation and spatial memory of young C57BL/6J mice.”, Lin et al., investigated the effects of moderate stress in vivo for one, two and four weeks. I found the manuscript to be well-written and below, the authors can find my comments:
Minor issues:
1. The abstract must be resized (200 words maximum)
2. Lines 96-97: The acronyms must be declared
3. Lines 90-94: How did the authors choose this protocol for inducing CMS? Is there any reference?
4. The authors must provide the references for all behavioral tests that they applied and they must specify the parameters measured within the tests.
5. Figure 1G: Please refer to open arms instead of novelty arm on the Y-axis of the graphs
6. Lines 240-242: OFT and EPM are not suitable for measuring depressive-like behavior. It is used frequently for measuring the degree of anxiety and locomotor activity. The authors must include anxiety in this results section.
7. Line 67: The authors must clearly mention the aim of the study. (in both abstract and introduction)
Major issues: None detected
As a recommendation, the beneficial stress is more and more referred to as "eustress".
Reviewer 3 Report
I have received the research article "A moderate duration of stress promotes behavioral adaptation and spatial memory of young C57BL/6J mice" by Lin et al, for evaluation. In their study, the authors have investigated the time-dependent effect of Chronic Mild Stress (CMS) on behaviors and cognitive performance, and synaptic plasticity biomarkers in mice hippocampus. The study is well planned and experiments are executed nicely.
Given how systematic and thorough the work has been conducted I find some comments regarding the manuscript.
Major comments:
1- Authors have used multiple approaches to induce the CMS in mice. Interestingly, CMS produced the biphasic response on the behaviors and cognitive performance and synaptic plasticity biomarkers in mice hippocampus at different time points. However, the authors missed evaluating the level of stress in the mice. I would suggest authors measure the cortisol level in the blood (stress indicator) at different time points.
2- In the GABA-immunoreactivity experiments, authors have quantified the anti-GABA positive neurons in mice. The authors should use a dual immune staining method [such as anti-GABA and anti-neuronal nuclei marker (anti-NueN)] for proper neuronal counting.
3- In the VGluT1-immunoreactivity experiments, authors have provided the result in the form of IOD/Area. First of all, authors should abbreviate the IOD. Second like GABA-positive neurons counting, authors should also count the VGluT1-positive neurons.
4- Since stratum pyramidal neurons are very much susceptible to stress. I would like to suggest authors check the effect of CMS in stratum pyramidal neurons.
Minor comments:
1- Is the paper title expressing the overall findings?
2- Authors should follow the same trend of data representation (please see graph axis title of Fig 2 A&B and Fig 3 A&B)
3- In the figure legend of Fig 2 and Fig 3, the authors should mention the number of mice (n=x) and number of observations per mouse (number of sections used) used in the experiments.
Round 2
Reviewer 3 Report
I have received the revised version of research article "A moderate duration of stress promotes behavioral adaptation and spatial memory of young C57BL/6J mice" by Lin et al, for evaluation.
Authors have tried to addressed my comments theoretically and they promise to adapt these comment in their future research. However, these comments are required especially for present study. My comments are mentioned below:-
My Comments
1- Since the study is showing the biphasic response in time dependent manor therefore authors should justify with experiment (stress indicator) that they are really inducing stress at both time point.
2-In the VGluT1-immunoreactivity experiments, authors expressed their limitation to count the cells.
Therefore I would suggest to present the zoom image (inset) of each panel.
Next, authors should also mention in the result section that "VGluT1-positive neurons were quantified by adapting semi quantitative method (IOD measurement)".
Authors should also mentioned in the method section that how they have substracted the background for IOD measurement.
